# Detection of β-Lactoglobulin by a Porous Silicon Microcavity Biosensor Based on the Angle Spectrum

**DOI:** 10.3390/s22051912

**Published:** 2022-03-01

**Authors:** Lanlan Bai, Yun Gao, Jiajia Wang, Tuerxunnayi Aili, Zhenhong Jia, Xiaoyi Lv, Xiaohui Huang, Jie Yang

**Affiliations:** 1School of Physical Science and Technology, Xinjiang University, Urumqi 830046, China; bll2019@stu.xju.edu.cn (L.B.); gyun@stu.xju.edu.cn (Y.G.); 2School of Information Science and Engineering, Xinjiang University, Urumqi 830046, China; wjjxj@xju.edu.cn (J.W.); hxhdemail@sina.com (X.H.); 3The Key Laboratory of Signal Detection and Processing, Xinjiang Uygur Autonomous Region, Xinjiang University, Urumqi 830046, China; lvxiaoyi@xju.edu.cn; 4School of Life Science and Technology, Xinjiang University, Urumqi 830046, China; tusnay@stu.xju.edu.cn (T.A.); yangjie234@xju.edu.cn (J.Y.); 5School of Software, Xinjiang University, Urumqi 830046, China

**Keywords:** porous silicon microcavity, carbon quantum dots, angular spectrum

## Abstract

In this paper, carbon quantum dot-labelled β-lactoglobulin antibodies were used for refractive index magnification, and β-lactoglobulin was detected by angle spectroscopy. In this method, the detection light is provided by a He-Ne laser whose central wavelength is the same as that of the porous silicon microcavity device, and the light source was changed to a parallel beam to illuminate the porous silicon microcavity’ surface by collimating beam expansion, and the reflected light was received on the porous silicon microcavity’ surface by a detector. The angle corresponding to the smallest luminous intensity before and after the onset of immune response was measured by a detector for different concentrations of β-lactoglobulin antigen and carbon quantum dot-labelled β-lactoglobulin antibodies, and the relationship between the variation in angle before and after the immune response was obtained for different concentrations of the β-lactoglobulin antigen. The results of the experiment present that the angle variations changed linearly with increasing β-lactoglobulin antigen concentration before and after the immune response. The limit of detection of β-lactoglobulin by this method was 0.73 μg/L, indicating that the method can be used to detect β-lactoglobulin quickly and conveniently at low cost.

## 1. Introduction

β-lactoglobulin (β-LG) is the most important protein in ruminant whey. The main allergen in milk is bovine β-LG, which is a foreign protein acquired by infants and young children. Moreover, β-LG is not easily decomposed by chymotrypsin and pepsin and can retain its high immune characteristics after digestion and absorption by the human body. Therefore, bovine β-LG is considered the most important allergen in milk [1]. Even at low concentrations, bovine β-LG can cause allergic reactions, resulting in symptoms such as gastrointestinal changes, atopic dermatitis, asthma, angioedema, or chronic cough [2]. Hence, detecting bovine β-LG is very important.

Porous silicon (PSi) is a kind of nano-silicon material [3], which has been generally applied in the territory of sensors [4,5] due to its own merits [6,7,8]. Currently, PSi sensors are classified into two main types: PSi chemical sensors [9] and PSi optical sensors [10]. Chemical substances [11,12] and biological substances [13,14] can be detected by these two types of PSi sensors. This study is based on the PSi optical biosensor. PSi optical biosensor has two detection mechanisms. One is based on using refractive index changes caused by biological reactions to detect changes in reflection spectrum or angle spectrum for biological detection [15]. Another detection mechanism is to use changes in fluorescence degree generated by fluorescent markers in reactants before and after biological reaction for biological detection [16,17]. This study is concerned with the sensing mechanism based on refractive index variation. When comparing the two detection mechanisms, the fluorescence detection method is more sensitive, but the fluorescence method requires labeling, while the refractive index method does not.

Currently, PSi optical biosensors with a variety of structures have been reported, most notably Bragg mirrors [18,19], microcavities [20,21] and surface gratings [22]. Among them, the porous silicon microcavity (PSM) sensor is a one-dimensional photonic crystal with defect states. The PSM sensor has a high-detection sensitivity, and its reflection spectrum at the defective peak has a tall transmittance, narrow half-width and other excellent optical features [23,24]. Microcavities are used as optical bandpass filters and interferometers in optical spectra. In these applications, cavity resonances are used to selectively transmit wavelengths in the spectral band, while angular cavity resonance-dependent or variable spacer layers are used to tune wavelengths [25]. Xiaoyi Lv et al. successfully used a PSM device, first fixing the antibody on the PSM device and then detecting the antigen by using an antigen–antibody specific reaction. According to the experimental results, the redshift of the PSM biosensor reflection spectrum increases with increasing antigen concentration [26]. Based on the PSM device, Rui Zhou et al. used angle spectroscopy to detect hybridization responses between DNA. The quantum dots marked target DNA, and the QDs play a role in refractive index amplification. The detection limit of target DNA by angle generalization method is 35.92 pM [27].

Carbon quantum dots (CQDs) have a broad excitation spectrum, a narrow emission spectrum, a large Stokes displacement, and good optical stability, allowing for long-term observation of labelled objects [28,29,30]. Since CQDs do not contain heavy metal elements, they do not have the high toxicity of inorganic semiconductor quantum dots [31,32].

In this paper, CQDs labeled β-LG antibody was used to amplify the refractive index of reactants, and angle spectrum detection method was used to detect the angle changes caused by immune reactions between CQDs labeled β-LG antibody and β -lactoglobulin antigen in different concentrations in PSM device. Firstly, β-LG antigen is fixed on the PSM device, and He-Ne laser (wavelength is the same as that of PSM device) is used to incident on the surface of PSM device after collimating beam expansion. In this case, the incident light with the same wavelength as the PSM device is incident to the surface of the PSM device after collimating expansion. The minimum reflected light intensity can be obtained at an angle denoted as θ_1_. When the CQDs-labeled β -lactoglobulin antibody is added to the holes in the PSM, the CQDs-labeled β-LG antibody will react with the β-LG antigen in the microcavity device, and the minimum reflected light intensity can be found again at an angle denoted as θ_2_. At this point, the angle change caused by the immune reaction of antigen and antibody can be recorded as ∆θ = θ_2_ − θ_1_. From this the angle change caused by the immune response can be calculated.

## 2. Experimental Principle

The PSM consists of two perfectly symmetrical Bragg mirrors (Bm) on the top and bottom, as well as a defect layer in the middle. The optical thickness between the dielectric layer and the defect layer satisfies the following relation:*n_H_d_H_* = *n_L_d_L_* = *λ_C_*/4(1)
*n_C_d_C_* = *λ_C_*/2(2)
where $\;n_{H}$, $n_{L}$, and $n_{C}$ delegate the refractive index of the high and low refractive index layer and the defect layer, respectively; $d_{H}$, $d_{L},$ and $d_{C}$ delegate the physical thickness of the high and low refractive index layer and the defect layer, respectively; and $\lambda_{C}$ delegates the wavelength of the resonant peak of the defect state.

According to Formulas (1) and (2), the theoretical reflection spectrum of PSM device is simulated by transfer matrix method. The PSM device is designed with 25 layers and its central wavelength is 633 nm. It is assumed that the thickness of the high and low refractive index layer of PSM device is 104 nm and 131 nm, respectively. The corresponding refractive index is 1.52 and 1.21, respectively. The thickness of the intermediate defect layer is 262 nm, and the relative refractive index is 1.21. Figure 1 shows the experimental device diagram of this study. After the He-Ne ( λ =633 nm , same as the set PSM device center wavelength) laser emits light, it collimates and expands the beam through the polarizer P, two lenses L_1_ and L_2_ and the aperture. The resulting beam irradiates to the surface of PSM device after passing through the aperture, and the reflected light intensity is received by detector D.

The central wavelength λ of PSM device is related to the effective refractive index n of PSM device, and its λ will change with the change of n. When the incident angle of the luminous source is changed so that the luminous source is incident obliquely, the reflection spectrum shifts correspondingly. On the basis of the above principle, the angle spectrum detection way is put forward. We chose a He-Ne laser whose wavelength is similar to the central wavelength of the PSM device as the luminous source. When the incident light of He-Ne vertically incident on the PSM surface, the reflected light intensity is smallest. When PSM is functionalized, the central wavelength of its PSM device will variation. The smallest value of the reflected luminous intensity can be gained by deflecting the incident light at a certain angle before the occurrence of the immune and the angle at this point is denoted as θ_1_. When the immune response occurs in the PSi hole, the effective refractive index of the PSi device variations, and the smallest value of luminous intensity can be gained by deflecting the incident light at a certain angle θ_2_. According to the angle variation (Δθ = θ_2_ − θ_1_), the refractive index variation Δn lead to the immune response in the PSM device which can be gained to realize biological detection.

Angle spectrum method is based on the change of angle ∆θ corresponding to the incident angle of minimum reflected light intensity before and after biological reaction. The refractive index variation ∆n can be obtained by measuring the angle variation of incident laser ∆θ to achieve the purpose of biological detection. Reflectance spectrometry is to use reflectance spectrometer to detect the redshift of the reflectance spectrum before and after the biological reaction. Compared with the reflection method, the angle method is a spectrometer-free method, only needing a laser, detector, goniometer, and some optical devices to complete detection and the total price of these instruments is only thousands of dollars. Whereas when using reflection spectroscopy, there is a need for spectral equipment and this equipment is tens of thousands of dollars. Therefore, the angle spectrum method is a low-cost detection and a spectrometer-free detection.

## 3. Materials and Methods

### 3.1. Preparation of the PSM

The PSM was fabricated by electrochemical etching with a single-slot anode using p-type boron-doped monocrystalline silicon (<100> crystal orientation, resistivity 0.03–0.06 Ω·cm, thickness of 400 ± 10 μm, Tianjin Semiconductor Technology Research Institute, Tianjin, China). Ahead of preparing the PSM device, the silicon pellet was carved up 2 cm × 2 cm squares, soaked in acetone (Tianjin Zhiyuan Chemical Reagent Co., Ltd., Tianjin, China), methanol (Tianjin Zhiyuan Chemical Reagent Co., Ltd., Tianjin, China) and deionized water in turn, and placed in an ultrasonic instrument for 20 min to remove residual surface impurities. After cleaning, the silicon wafer was removed and dried under nitrogen. Using Labview (National Instruments), the current densities of the high and low refractive index layers and the middle microcavity layer were set at 40 mA/cm^2^, 90 mA/cm^2^, and 90 mA/cm^2^, respectively, and the corrosion time was 2 s, 1.7 s, and 3.4 s, respectively. In order to ensure the uniformity of corrosion, during the corrosion process, the formation of a layer of PSi should be separated by 3 s, and the whole process can only be completed in a ventilated environment. After the corrosion process is completed, the corroded PSi was washed several times with deionized water and dried in a nitrogen environment. The reflection spectrum of the PSM device was measured by a spectrometer with a resolution of 0.1 nm (Hitachi U-4100, purchased from Hitachi, Japan) and compared with the reflection spectrum calculated by the transfer matrix method, as shown in Figure 2. 

Figure 2 is a comparison of the experimental results and theoretical simulation results of the reflection spectra of the PSM device prepared for the experiment. The central wavelength of the prepared PSM device is 633 nm, which is consistent with the theoretical simulation. Due to the PSM’s surface fluctuations, the reflectance of the reflection spectrum of the PSM device decreases in the band gap but increases in the defect state, and thus, the half-width of the defect state increases [33]. As a result, there are some differences between the experimental results and the theoretical simulations.

Figure 3 presents an SEM image of the PSM surface device prepared for the experiment. Its pore size is approximately 30 nm, which allows the CQD-labelled β-LG antibody to enter smoothly and perform biological detection. Each layer of the Bm and the defect layer in the middle of the PSM device may be distinctly seen in the cross-sectional diagram.

### 3.2. Functionalization of the PSM

Because the newly prepared PSi has Si-H bonds on its surface, it oxidizes easily when exposed to air. Therefore, the PSi needs to be oxidized at 60 °C. The oxidation solution was 30% hydrogen peroxide (Tianjin Zhiyuan Chemical Reagent Co., Ltd., Tianjin, China), and the oxidation time was 3 h to make the surface of the PSM device have the SiO_2_ layer. After removal, the device was repeatedly rinsed with deionized water to remove hydrogen peroxide from the surface of the microcavity device. Then, it was silanized and placed in 5% 3-aminopropyl triethoxysilane (Aladdin Reagent Co., Ltd. Shanghai, China) for an hour to generate amino groups on the PSM surface device. Next, the PSM device was removed and washed with deionized water several times, blown dry indoor temperature, and dried for ten minutes in a vacuum (DZF-6050, Shanghai Yiheng Scientific Instrument Co., Ltd., Shanghai, China) drying oven at 100 °C. For the successful addition of the β-LG antigen, the silanized samples were dipped in glutaraldehyde solution (at a concentration of 2.5% Aladdin Reagent Co., Ltd., Shanghai, China) for an hour, followed by a phosphate-buffered solution (PBS, PH = 7.4), and finally washed with deionized water. Each functionalization stage in a process of the PSM devices must be measured by reflection angle spectroscopy to verify successful functionalization. Figure 4a shows the angle spectrum contrast diagram for the PSM functionalization, Figure 4b comparison of the reflection spectra of functionalized PSM. Figure 4 indicates that each stage in a process of functionalization was completed successfully.

### 3.3. Preparation of the Antibody Serum 

Both the β-LG antigen and the β-LG antibody used in this study were obtained from the College of Life Science and Technology, Xinjiang University. Preparation method of antibody serum: Blood samples were collected from 8 rabbits to obtain pre-immunization serum. Milk β-LG standard product was mixed with fredrin complete adjuvant at a specific concentration to carry out the primary immunization. After the end of the primary immunization, the antigen immune dose was reduced, and the second immunization was used to strengthen the immunity. At the end of the third and fourth enhanced immune response time, venous blood was collected to obtain antibody serum purification for use.

### 3.4. Preparation of the β-LG Antigen and Quantum-Dot-Modified β-LG Antibody

After the PSM was functionalized, 50 μL of β-LG antigen at concentrations of 0.5 μg/L, 1 μg/L, 5 μg/L, 10 μg/L, and 15 μg/L was dropped onto the surface of the PSM device by microtransfer and put into an incubator at 37 °C for about two hours. After the β-LG antigen was inserted into the PSM device, it was washed with PBS, washed with deionized water, and then dried in nitrogen. Next, the PSM device was immersed in a bovine serum albumin solution (4%) and put in an incubator 37 °C for an hour. Subsequently, the device was rinsed several times with PBS to remove excess bovine serum albumin. Finally, the device was rinsed several times with deionized water and dried under nitrogen.

CQDs label antibodies by covalently binding. In this paper, carbon quantum dots (CQDs) were purchased from Nanjing Genase New Materials Co., Ltd., Nanjing, China. The particle size is 10 nm and the fluorescence peak is located at 627 nm. CQDs were diluted from 10 g/L to 0.1 mg/L, and then, 0.01 M 1-ethyl- (3- dimethylaminopropyl) carbodiimide hydrochloride (EDC) and sulfo-N-hydroxysulfosuccinimde (NHS) were added to activate the CQDs. EDC and NHS were obtained from the Tianjin Zhiyuan Chemical Reagent Co., Ltd., (Tianjin, China). After an hour at indoor temperature, the required concentration of 50 μL of β-LG antibody was added, and light shock was conducted for ten hours. The CQDs and β-LG antibodies were fully connected before being centrifuged in a 5000 r/min centrifuge for ten minutes. The CQD-labelled β-LG antibody was dropped onto the surface of the PSM device with the β-LG antigen fixed by a pipetting device and incubated at 37 °C for two hours. Then, the PSM device was removed and the antibody was rinsed several times with PBS and then with deionized water to remove CQDs that failed to connect and CQDs-modified β -lactoglobulin antibodies that failed to enter the hole for an immune response. The process of biosensor preparation is shown in Figure 5.

## 4. Results and Discussion

Figure 6a is a high-power transmission (Xinjiang University Test Center) electron microscope image of CQDs. To demonstrate the successful coupling of the CQDs with the β-LG antibody, fluorog-3-21-TCSPC (Hitachi F-4600, Japan wavelength resolution 0.2 nm) was used to measure the fluorescence peak of the CQDs before and after coupling with the β-LG antibody. Figure 6b presents that the fluorescence emission peak of CQDs without the β-LG antibody was 627 nm, and that of CQDs with a β-LG antibody concentration of 10 μg/L was 630 nm. The redshift was 3 nm, indicating that the CQDs and β-LG antibody were successfully conjugated [34].

After functionalization and the addition of the antigen and antibody, the central wavelength of the PSM device changes, leading to a dissimilarity with the wavelength of the incident light. To make the central wavelength of the PSM conform to the incident photosource, we used the oblique incidence method. After addition of the β-LG antigen and β-LG antibody and the immune response, the angles of the minimum reflected luminous intensity of the PSM device before the immune response and after the immune response were sought out the oblique incidence way and denoted as θ_1_ and θ_2_, respectively. The variation in the angle before the immune response and after the immune response was Δθ = θ_2_ − θ_1_. In order to precisely seek the location of the smallest luminous intensity, the luminous intensity was recorded for each 1’ variation in the approximate angle of the location of the smallest luminous intensity, as shown in Figure 7a. It can be seen from Figure 7b that the β-LG antigen and the CQD-labelled β-LG antibody are connected to PSM after a redshift of reflection spectrum, which proves that immune reaction occurs in PSM.

The concentrations of the β -lactoglobulin antigen and CQDs antibody used in this study were 30 μg/L and 100 μg/L, respectively. To achieve complete reaction at different concentrations of β-LG antigen, we applied CQDs antibody at a concentration of 100 μg/L. Figure 7a shows that after adding the CQD-modified β-LG antibody to the PSM device with fixed β-LG antigens, the angle spectrum changed by 5.33°. The angle of the PSM device clearly changed, indicating an immune reaction between the β-LG antigen and its corresponding antibody, which increased the effective refractive index of the PSM device, resulting in a redshift of the angle spectrum.

To detect whether there were CQDs that had either not been washed out or were not in the holes of the PSM devices, a fluorescence spectrometer was used (Hitachi F4600, Hitachi Company, Tokyo, Japan, with a wavelength resolution of 0.1 nm). Residual CQDs in the PSM devices are attributed to two factors. The first is that CQDs can fail to conjugate with the β-LG antibody successfully. The second is that CQDs successfully conjugate with the β-LG antibodies, but the immune reaction between the CQDs and β-LG antigen fails. Because the surface groups of the experimentally prepared PSi are closed after functionalization, the CQDs unable immediately link with the PSi pore wall. Figure 8 shows that the excitation wavelength is 530 nm, the excitation voltage is 700 V, and the slit width is 5 nm. The black curve indicates CQDs that were dropped immediately onto the PSi, and the red curve manifests CQDs that were labelled with the β-LG antibody and then conjugated with the functionalized PSi. The results show that CQDs unable, pass into the PSi holes, if the CQDs are not successfully conjugated with the β-LG antibody.

The concentrations of the β-LG antigen fixed in the PSM device were 0.5 μg/L, 1 μg/L, 5 μg/L, 10 μg/L, and 15 μg/L, and an immune reaction occurred with the CQD-labelled β-LG antibodies. The redshift angles Δθ were 0.33°, 0.53°, 1.38°, 2.70°, and 4.55°. The fitting results are shown in Figure 9. The linear equation is Y = 0.283X + 0.148 where X delegates the concentration of the β-LG antigen (μg/L) and Y delegates the redshift angle Δθ before and after the immune reaction. The slope is 0.283, and the linear correlation coefficient is R^2^ = 0.99. In this study, calibration curves were obtained through 5 repeated experiments.

Since the PSi surface device has ups and downs, it will lead to errors in the experimental measurement results, so we use the 3 σ rule to compute the experimental results. Adopting the alike PSi blank sample, the location of the smallest reflected luminous intensity was measured ten times, and the standard deviation σ was obtained. The experimental result was σ = 0.069, indicating that the detection limit was 0.73 μg/L.

As for the sensitivity, we can conclude the following theoretical analysis. When the refractive index of the porous silicon microcavity increases by 10^−4^, the change of the angle ∆θ=0.046° can be calculated by transfer matrix method. The resolution of the spectroscopic equipment we are currently using is 1’, i.e., 0.0167°. Considering the dispersion and interface roughness of porous silicon devices, the actual minimum angle variation we can resolve is 0.2°. When the refractive index increases by 5 × 10^−4^, we calculate the change of the angle ∆θ = 0.22°. Therefore, it is completely possible to measure using the angle spectrum method [15]. Therefore, according to the fitting curve in Figure 9, we can calculate that the sensitivity of the sensor is 0.28°/(μg/L).

Immunological assays have previously been used to detect β-LG. Reference [35] reported β-LG detection limits of 1.96 μg/L by (Enzyme-Linked Immuno Sorbent Assay) ELISA, respectively. In reference [36], the detection limit was 0.2 μg/L by immunochromatography. In references [37,38], the β-LG detection limits were 0.8 μg/L and 164 μg/L, respectively, by electrochemical immunoassays. Compared with that of the methods used in the above references, the β-LG detection limit for our method was 0.73 μg /L, which was the same order of magnitude. In addition, our detection method has the benefits of a short detection time and low detection cost.

## 5. Conclusions

In this paper, a β-LG antibody was successfully labelled by CQDs, and the refractive index of the reactants was magnified. The immunoreactivity of the β-LG antigen and CQD-labelled β-LG antibody at different concentrations was tested by an angle spectrum assay applying a spectroscopy-free device. The experimental results showed that the angle changed linearly with the β-LG antigen concentration before and after the immune response. The β-LG detection limit for this method was 0.73 μg/L. This method can be used to detect β-LG quickly and conveniently at a low cost.

## Figures and Tables

**Figure 1 sensors-22-01912-f001:**
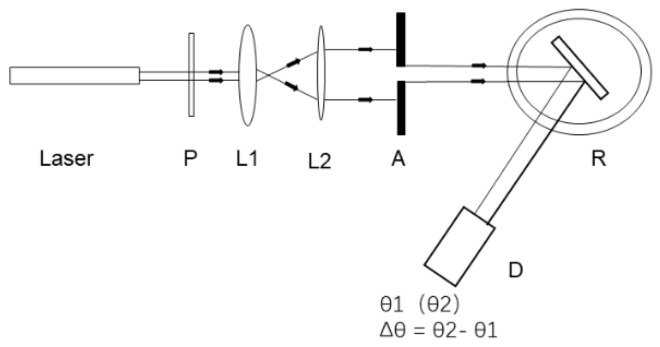
Schematic diagram of experimental device.

**Figure 2 sensors-22-01912-f002:**
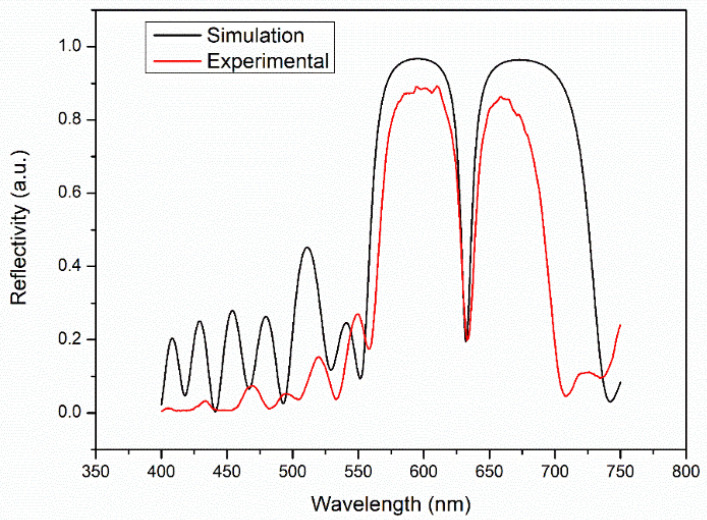
Comparison of PSM reflection spectra from the experimental results and theoretical simulations.

**Figure 3 sensors-22-01912-f003:**
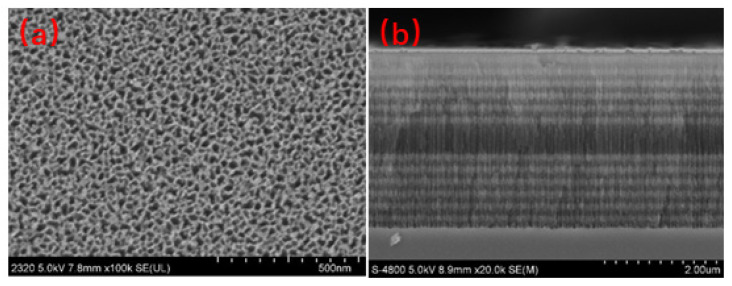
(**a**) SEM surface view and (**b**) SEM cross-sectional view of the PSM.

**Figure 4 sensors-22-01912-f004:**
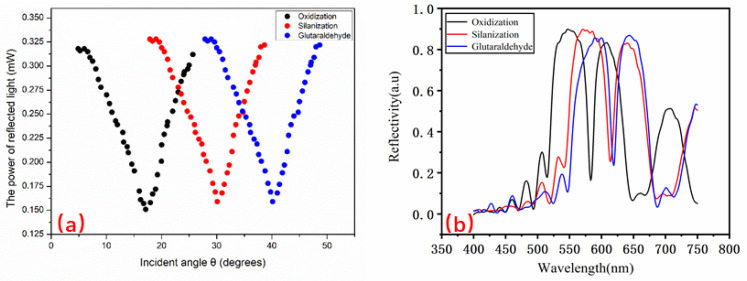
(**a**) Angle spectrum contrast for the PSM functionalization; (**b**) Comparison of the reflection spectra of functionalized PSM.

**Figure 5 sensors-22-01912-f005:**
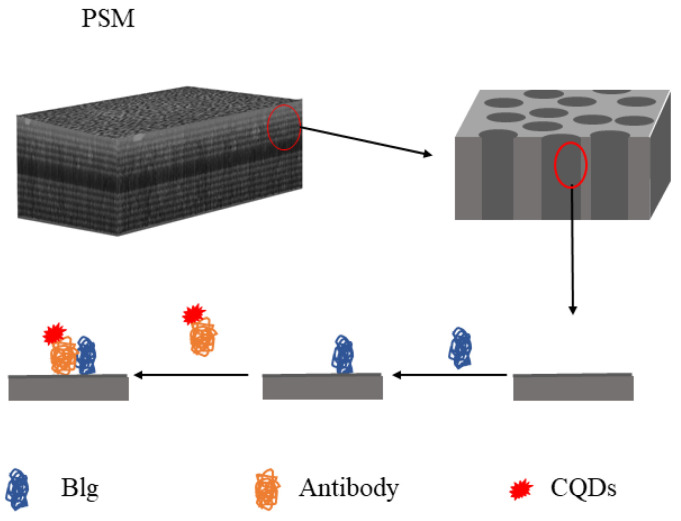
CQDs labeling of CQD-linked β-LG antibody and the process of immune response.

**Figure 6 sensors-22-01912-f006:**
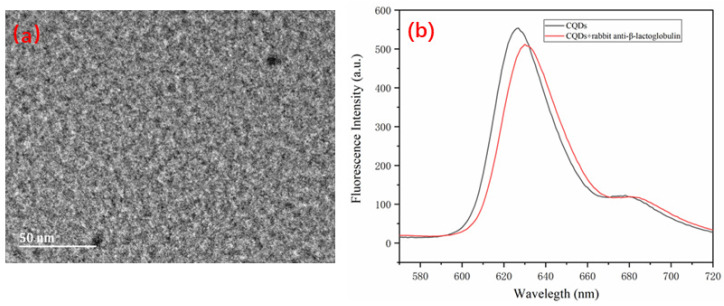
(**a**) High-power transmission electron microscope image of carbon quantum dots; (**b**) Fluorescence spectra of the CQDs before and after conjugation with the β-LG antibody.

**Figure 7 sensors-22-01912-f007:**
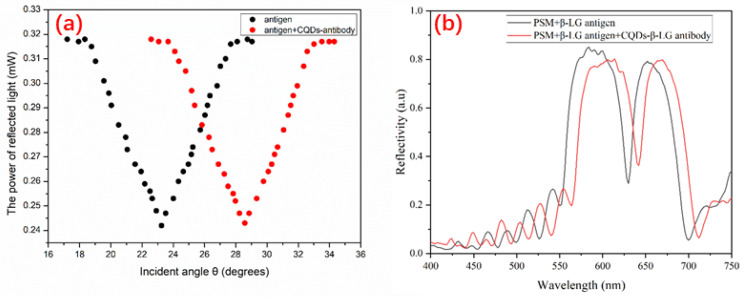
(**a**) Angle spectrum before the immune response and after the immune response. The black curve expresses the angle spectrum before the immune response, and the red curve represents the angle spectrum of the CQD-conjugated β-LG antibody; (**b**) Contrast of reflection spectra before and after connecting the β -LG antigen and CQDs-β-LG antibody in PSM.

**Figure 8 sensors-22-01912-f008:**
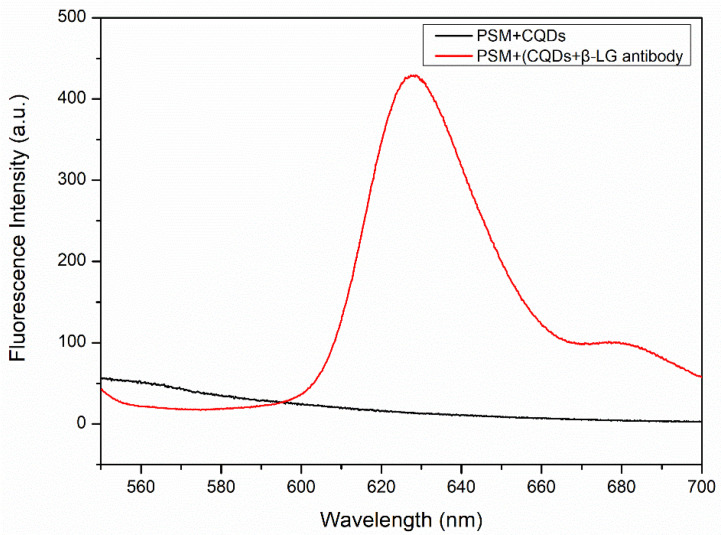
The red curve represents the fluorescence spectra of 100 μg/L CQDS-linked β-LG antibody after addition to a functionalized PSM device, and the black curve represents the fluorescence spectra of QDs after addition to a functionalized PSM device.

**Figure 9 sensors-22-01912-f009:**
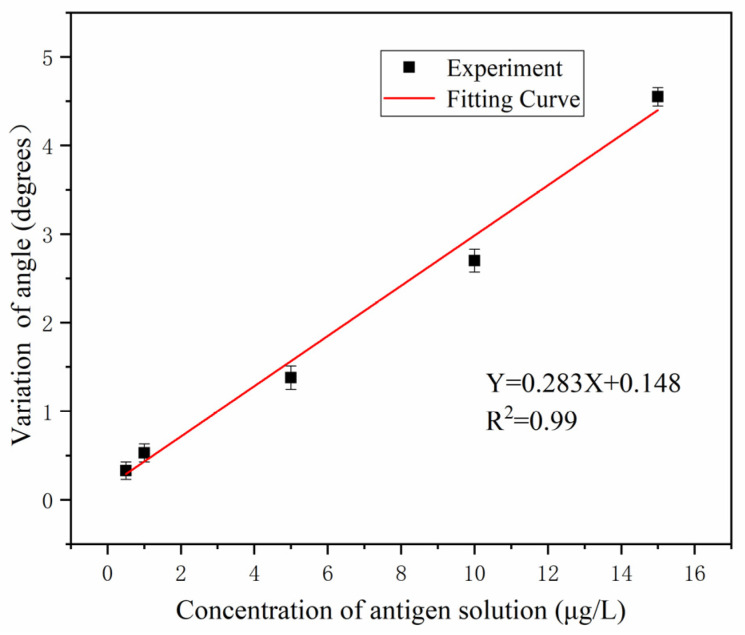
The relationship between different concentrations of the β-LG antigen and the angle spectrum redshift.

## Data Availability

Not applicable.

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
