# Peer review of "Detection of β-Lactoglobulin by a Porous Silicon Microcavity Biosensor Based on the Angle Spectrum"

_sensors, 2022, doi:10.3390/s22051912_

Round 1
Reviewer 1 Report
This submission is related to the detection of β-lactoglobulin, a well-known milk allergen, using a porous silicon microcavity (PSM) formed by a porous defect layer trapped between two Bragg mirrors. A selective detection takes place between the antigen attached to the porous silicon surface and the β-lactoglobulin antibody. The detection is then based on the measurement of angle-resolved reflectance spectra amplified by carbon quantum dots (CQD) attached to the antibody.
Porous silicon has been widely used as biosensing platform, often in combination with antibody-antigen specific recognition. The detection of β-lactoglobulin has gained interest gradually over the last 30 years even if it remains at a moderate level. More recently, several label-free techniques are competing in this field for low costs and fast response times.
With respect to these two points, the submission is well aligned and is a topic worth of investigation although not satisfactory in the present form.
As a first general comment, the overall structure of the article is not clear and must be revised to provide a better, more readable structure. The Results section should be subdivided for the characterization of the PSM, the characterization of the CQD conjugation and the penetration in the porous material, and then a final section on the detection itself. Moreover, all methods used for the characterization should be detailed in the Methods section.
As a second general comment, the conclusions are based on measurements performed with a single device. The claimed gain for the limit of detection with respect to the state of the art must be confirmed by reproducing these results on a larger set of devices prepared using the same method.
As a third general comment, the state of the art is missing the most recent improvements in electrochemical sensors used for the detection of β-lactoglobulin with much lower detection limits. Analysis of real samples would also have been an added value to the submission, and clearer comparison of the overall cost and time to other techniques would be needed for fairness.
Specific comments:
- Try to avoid the repetition of the same idea or the same terms in the first paragraph. For example, lines 32-33 and lines 35-36 express the same idea. Other examples, the word “Therefore” appears twice in the same paragraph, try replacing one of the instances by a synonym.
- Acronyms must be redefined in the main text, even if already defined in the abstract.
- Write all units it the same correct way (degree Celsius should be °C, percent should be %, etc.).
- End of the introduction: give some more details about the structure of your article to guide the reader.
- Either gather all the purchased materials in a Materials section and specify the origin of each material, or specify the origin for all Materials in the text, but do not do it only sporadically.
- Verify the titles of your sections (3.1 is not a correct title, missing title at 3.1.1, etc.).
- Can you provide a source for your statement lines 50-51 “the fluorescence detection method is 50 more sensitive”?
- Figure 2 and 3 are results.
- Line 140: pore size and not aperture size.
- Lines 150-151: “To make the surface of PSM device have SiO2 layer” is not a full sentence.
- Preparation of the antibody serum: make this paragraph a subsection.
- Lines 191-192: not a full sentence.
- Lines 197-198: should be in the Materials & Methods section.
- Lines 199-201, 209-216, 236-237: all methods should be described in the Materials & Methods section.
- Lines 247: what is the error on those values? It is not clear based on how many samples these values where calculated.
---End of the comments.
Author Response
Our answers to your questions can be found in the uploaded document.

Reviewer 2 Report
Bai et al. report a study in which the photonic features of a functionalized porous silicon microcavity are exploited to implement a beta-lactoglobulin biosensor, which relies on an angle-resolved, single-wavelength evaluation method. The limit of detection of this system results comparable to that of other assays reported in the literature. The topic can certainly be of interest for any researcher working in the fields of food safety and nanotechnology for healthcare. Indeed, porous silicon represents a technology with a sufficient degree of maturity to allow the development of reliable and marketable devices.
Despite this good premise, I believe that this manuscript exhibits critical flaws and voids. My considerations, which involve both the experimental (design and implementation of the conducted research) and the writing accuracy viewpoints, are listed afterwards.
Experimental issues:
1. Why was not angle spectroscopy used for the CQDs (without antibodies) infiltration test? Also, what is the point of using the red-emitting CQDs as RI amplifier if a 633 nm excitation is used in the sensing experiments?
2. At least one more characterization should be provided to verify the correct conclusion of each step of the functionalization (e.g.: WCA, FTIR, reflectance…). Same applies for proving the immune interaction.
3. A DLS or TEM characterization should be given to confirm that the size of the CQDs-Ab complex does not exceed that of the pores of the functionalized PSi platform.
4. Which concentrations of bLG and CQDs-Ab does the red plot in Fig. 7a correspond to? Also, Fig. 7b is redundant as freestanding, it could be put as an inset.
5. What was the utilized CQDs-Ab concentration for the complete immune response experiment?
6. Over how many replicates was the calibration curve obtained? No statistics is evidenced (except for the linear R2).
7. No data are provided regarding the specificity/selectivity of the device. Since the authors claim to have developed a biosensor, tests with interferent molecules and other analytes should definitely be carried out (at least at a preliminary stage).
8. What do the authors mean by “blank” - Line 257?
Writing quality issues:
1. The introduction section could be extended and improved by better justifying the use of porous silicon for optical biosensing applications. I have read only few vague lines as motivation. Furthermore, the explanation of the sensing transduction at the end of the introduction could be directly moved in the experimental section.
2. The Materials and Methods section is rich in Figures (which should be part of the Results and discussion section) but lacks of relevant information. Only few reagents and instrumentations specifications have been reported.
3. No comment or accurate description has been provided for the optical setup illustrated in Figure 1. The paper needs to be as much accessible as possible even to differently specialized audience. Nevertheless, the word “device” is used to describe both the measurement setup for angle spectroscopy as well as the biosensing porous silicon chip. This is ambiguous and misleading (see next point).
4. (Critical point of the list) The English vocabulary and style are vague and inattentive throughout the whole paper, making it difficult to reach a clear comprehension of its content (few examples: what does the sentence in LL. 135-137 mean?; The sentence in LL. 235-237 is out of context; what do the authors mean by “the PSi’ surface device has ups and downs” - Line 255?).
5. How can a platform that requires a (totally non-specific) pre-immobilization of the target analyte (which may require complex raw samples purification procedures) be low-cost? How can the proposed device be practically used? It is not clearly explained.
Other minor issues that could be addressed (which, however, I always suggest to handle regardless the post-acceptance proofing) are the following:
1. Acronyms should be introduced before using them (e.g.: PSM, Bm). I would suggest avoiding doing this in the abstract, but only in the main text.
2. Sections and paragraphs should be checked (there is a “subsubsection” label – Line 115).
3. Despite µg/L and ng/mL are identical units, there is no point in using both notations.
Author Response

(The authors gave the same response as above.)

Reviewer 3 Report
In this work, authors have demonstrated the application of porous silicon microcavity biosensor in detecting β-lactoglobulin by a cost-friendly technique based on refractive index variation in reflection spectrum/ angle spectrum expressing biological responses. However, the present form of the manuscript is not free of errors, and the accordingly, the write up should be thoroughly revised based on the underlying comments:
- In the introduction part, technical language should be changed. For instance, the portion, ‘This study is based on PSi optical biosensors that have two detection mechanisms. One detection mechanism is based on using the refractive index changes caused by biological responses to detect variations in the reflection spectrum or angle spectrum for biological detection. The other detection mechanism is based on using the fluorescence degree variations produced by fluorescent markers in reactants before biological response and after biological response for biological detection.’ suggests that both of these detection mechanisms have been followed in this particular work. However, in reality, the present work has dealt with the sensing mechanism based on changes in refractive index.
- In the second para of the introduction, the authors stated that the fluorescence method requires labelling. However, in recent times, a number of nonconventional intrinsically fluorescent materials have been reported, wherein the labelling procedure has been avoided.
- In this communication, authors have employed a β-LG antibody labelled with CQD. Labelling is cumbersome but the authors have adopted such levelling procedure-Why?
- Authors should have discussed the role of microcavity while discussing the sensing by PSM biosensor.
- For better understanding of the readers, authors should have marked θ1 and θ2 and Δθ in figure 1.
- During functionalization of PSM, there is a need for two stage washing, i.e., washing by deionized water and anhydrous ethanol in two stages. Authors should include reasons behind such two stages washing.
- Authors should have indicated the pH of phosphate buffer solution.
- Authors should cite a reference in relation to the confirmation of conjugation of CQDs with β-LG antibody by red shift of only 3 nm. In fact, such small variation may occur while repeating the experiments.
- In figure 9, the correlation coefficient (R2) value of 0.98, which seems to be insufficient and needs to be improved.
- Some full forms of abbreviations, i.e., ELISA, EDC, and NHS, are missing.
- There are some unwanted grammatical mistakes, typos errors, and inconsistency throughout the communication.
Finally, the present form of the manuscript needs through revision before it can be processed further for the next stages.
Author Response

(The authors gave the same response as above.)

Round 2
Reviewer 1 Report
Dear Authors,
Thank you for addressing the specific comments through your responses and to propose a revised manuscript that is somehow an improvement with respect to the first one.
However, your responses are missing addressing the three general comments that have led to request a major revision. Please read them back in the first reviewer's report as they won't be repeated here.
This revised version is still badly structured, with results in the Methods section and vice versa, and results are still missing to support the conclusions particularly with respect to the comparison between the angular spectrum measurement technique and the simple reflectivity measurement. Figs. 8 and 9 are showing both, but the text then only relates to the angle without much explanation, a comparison of both techniques and with literature data must have been performed here.
Line 51: "detection mechanisms" instead of "etection mechanisms"
§ from line 41 to 55 has conflicting information: first it details that this study describes two detection mechanisms, and then end by stating that only one is used in reality. Please make this paragraph more understandable, or as said before perform actually the comparison of both. The article length can afford it.
Lines 82-83: "incidentially incident" makes no sense.
Fig. 1 legend is a bit vague and general.
Line 126: replace "A PSM was prepared" by "Preparation of the PSM".
Fig. 2 is actually a PSM characterization result, it would help to have a subsection dedicated to all your characterization results on the PSM, such as also line 153 to 158 and Fig. 3.
Fig. 4 should go to the Results section.
Line 197, put in the incubator at ?? degrees Celsius; once the temperature is set please use °C for units.
Lines 218-219 should appear in the Materials section; lines 221-223 should appear in the Methods section.
Fig. 6: where are the 10 nm QD in this image? It only looks like noise, no visible appearance of the particles.
Fig. 7 in the legend "antibody".
Fig. 9: Did you quantify the red shift in the optical spectra? For which concentration is this graph? Did you also quantify the shift for different concentrations? It might be interesting to do so and compare to the literature. A calculation of the detection limit for this technique would be needed and check if it reaches levels similar to the angle spectrum one.
For consistency, please use the same units overall with respect to detection limit as there are ug/L and ng/mL appearing; although the conversion is simple, it would help the reader if all units are similar.
--- End of the comments.
Author Response
We very much appreciate the careful reading of our manuscript and valuable suggestions of reviewers. We have carefully considered the comments and have revised the manuscript accordingly.

Reviewer 2 Report
The manuscript by Bai et al. has improved after this round of peer-review and the authors have sufficiently addressed part of the initial issues. However, further refinements need to be fulfilled prior to paper acceptance:
1. The red fluorescence from CQDs is excited at 530 nm, i.e., using green light. At the same time, the PSM structure is specifically designed for interrogation at 630 nm. In this case, the emission property of the CQDs results irrelevant to the purpose of the paper. Indeed, a valid explanation of why the CQDs were used as RI amplifier in the sensing experiments is still missing (part 1 - question 1 of my previous report).
2. In my previous report, I asked for further characterization of the whole functionalization procedure. Despite the authors have appropriately added the reflectance data to validate the occurring of the immune interaction, the same analysis for each step (oxidation, silanization and glutaraldehyde conjugation) is still missing. An extension and integration of Fig. 4 would greatly improve the paper.
3. The authors are invited to include in the manuscript the data they appropriately reported in their response, namely the concentrations of bLG and CQDs-Ab corresponding to the red plot in Fig. 8 as well as the utilized CQDs-Ab concentration for the complete immune response experiment. Regarding this latter value, a short sentence explaining why they chose it should also be added.
4. The authors are also invited to insert the number of replicates for the calibration curve in the main text (which, also in this case, was appropriately reported in the response).
5. The absence of data regarding the specificity/selectivity of the device still represents a serious issue. The proposed detection methodology must be validated in this sense (at least at a preliminary stage) regardless what already exists in the literature.
6. The English language and style employed still suffers from relevant lacks. Here are few examples:
- The optical setup for angle spectra measurements is still not properly described (which optical components have been used and why? I understand them but any interested reader with a different research background would be in difficulty).
- The sentence “Due to the PSM’ surface fluctuations, the reflectance of the reflection spectrum of the PSM device decreases in the band gap but increases in the defect state, and thus, the half-width of the defect state increases [33].” (LL. 149-151) should be appropriately rewritten in accordance with the explanation provided by the authors in their response. Same applies for “the PSi’ surface device has ups and downs” (Line 285).
- The sentence “He-Ne laser […] is used to incidentially incident on the surface of PSM device after collimating beam expansion. In this case, the incident light with the same wavelength as that of PSM device is incidentially incident on the surface of PSM device after collimating beam expansion.” (LL. 80-83) should be appropriately rewritten.
- “From this the change in refractive index caused by the immune response can be calculated.” (Line 89) how? The details and outcomes of these calculations are missing.
7. (minor suggestion) The authors have sufficiently clarified the advantages of the angle spectrum method in the response. However, I believe that this point could be better highlighted in the main text too.
8. (minor suggestion) The figures are too many and could be merged into fewer, denser ones (e.g., Figs. 6-7, Figs 8-9). Also, enumeration of figures and their recalling in the text should be updated and carefully checked (there are two “Figure 9” labels).
Author Response

(The authors gave the same response as above.)

Reviewer 3 Report
The authors have considered all issues raised in review, and substantially modified the article as suggested. Accordingly, I am recommending for publication in its current form.
Author Response
We very much appreciate the careful reading of our manuscript and valuable suggestions of reviewers.
Round 3
Reviewer 2 Report
The authors have sufficiently addressed all the relevant points. Apart from a general check for language inaccuracies, this paper can be accepted for publication in Sensors.